# Health and Well-Being of Church Musicians during the COVID-19 Pandemic—Experiences of Health and Work-Related Distress from Musicians of the Evangelical Lutheran Church in Finland

**DOI:** 10.3390/ijerph19169866

**Published:** 2022-08-10

**Authors:** Tuire Kuusi, Satu Viertiö, Anna Helenius, Kati Tervo-Niemelä

**Affiliations:** 1Sibelius Academy, Uniarts Helsinki, 00100 Helsinki, Finland; 2Department of Public Health and Welfare, Finnish Institute for Health and Welfare in Finland (THL), 00271 Helsinki, Finland; 3Faculty of Philosophy, Department of Theology, University of Eastern Finland, 70211 Kuopio, Finland

**Keywords:** self-rated health, burnout, work engagement, well-being at work, church musician, gender differences

## Abstract

Earlier research has revealed contrasting gender results in standardized mortality ratios (SMR) for cancers and cardiovascular diseases of Finnish church musicians compared with the general population. In order to better understand the SMRs, our study examined gender differences in health and work-related experiences of church musicians with special focus on experienced stress and burnout on the one hand, and work engagement and mental well-being on the other. The data were collected by a questionnaire including both standardized measures and open-ended questions. Statistical methods (mostly χ^2^ tests) were used for examining gender differences in the measures, and the open-ended questions were analyzed using theory-driven content analysis. The two sets of data complemented each other. Analyses of the standardized measures showed that church musicians have more burnout and distress than the general population but the results were not gendered. However, the open-ended questions revealed clearly higher distress in females than in males. Based on the contrast between the measures and the open-ended questions, we raise the question about how well females who have distressing work can recognize the stress factors and change them, especially if distress becomes a “normal state”.

## 1. Introduction

In 2019 a surprising gender difference was found in the standardized mortality ratio (SMR) of Finnish church musicians. Although the SMR for both cancers and cardiovascular diseases was lower than the population average, the SMR among female church musicians was higher than among males in both disease classes. For cancer, the SMR for females was 0.90 (95% CI: 0.83–0.97) and for males 0.60 (95% CI: 0.54–0.67). The SMRs for cardiovascular disease were 0.75 (95% CI: 0.68–0.82) among females and 0.58 (95% CI: 0.54–0.64) among males [1]. The result contrasted with the population average, because at the time of the study, mortality from cardiovascular diseases in males was higher than in females [2,3]. Similarly, there were more cancer deaths at the population level in males than in females at the time of the study [4]. Why did the church musicians differ from the general population? In Finland, church musicians are highly educated, full-time professionals working in congregations in paid posts. Compared with the general population, the educational background most likely explains the lower SMR for both cancers and cardiovascular diseases in church musicians, but it does not explain the gender differences.

Even though the current work environment cannot explain deaths of earlier generations, the findings raised the question of the possibly gendered work environment of Finnish church musicians. An earlier study found that 31% of church musicians have so much work that they at least sometimes cannot cope with it, and that approximately 20% of them have daily or weekly feelings of exhaustion and cynicism. The mean rates for cynicism and inefficacy were higher than among the clergy and among the general Finnish population. Further, the mean rates for exhaustion were higher than among the clergy, but slightly lower than among general population. The study also showed an increase in the number of exhausted church musicians [5]. Thus far there is, however, no study on health issues nor gender differences of Finnish church musicians. Our aim and main research question was, hence, to examine what kind of gender differences can be revealed in church musicians’ health and work-related well-being. The supplementary research questions examine (1) how the church musicians experience their health; (2) what work-related distresses they have; and (3) what reasons they recognize for distress. To obtain a thorough view, we collected data using both standardized measures (quantitative data) and open-ended questions (qualitative data) concerning health and work-related experiences of the church musicians of the Evangelical Lutheran Church of Finland in 2020–2021. The project had a special focus on the time when the COVID-19 pandemic brought its own additional challenges to work by moving the normal events (e.g., services, group meetings, choir rehearsals) online or suspending them for an undefined time (e.g., visits to elderly people’s homes, day care centres, schools, etc.). One of the biggest changes was the need to learn to stream the services. Further, the work in which workers normally encounter each other and the members of the congregation, turned into work done from home using telecommuting equipment.

Since the data was collected by a questionnaire, we studied **subjective experiences of health (self-rated health)**, which might be different from health evaluated and measured by medical experts. The experienced health of the Finnish general population is frequently studied and reported by the Finnish Institute for Health and Welfare (THL) [6]. We used these same measures in our study and compared our results with two reports provided by the institute [7,8]. In the various sets of data and reports of THL, the experiences of health are often divided into physical health and work-related well-being. The division is not exclusive, since there is strong interaction between work-related well-being and health. Physical health issues are further divided into lifestyle (nutrition, exercise and rest) and health (diseases, symptoms, risk factors and ability to act) [7,8]. Many of these issues are connected to the level of education [7] (pp. 179–182), but since all our participants were educated as church musicians, we assumed that the degree of education is very similar (if not the same) for all participants and cannot be used as an explanatory factor. Further, paid employment, which is one determinant of well-being [9], was common for all participants of our study.

Considering work-related well-being and following Maslach [10], we investigated two opposite perspectives. On the one hand, we examined positive experiences, such as work engagement, positive psychological well-being and mental health, and on the other hand, psychological distress and burnout experiences. Below, we examine the aspects of work-related well-being and mental health that are relevant for our study. The measures are described in Methods.

The ideas of **work engagement** are based on the job demands–resources theory [11]. It has been described as a combination of work characteristics (working conditions and physical, psychological, social and organizational resources at work), the resources of the worker (e.g., personal characteristics and resources [12], control, self-efficacy, optimism, resilience), situations in work and in personal life [13], and job resources [14]. Traditionally it means positive well-being experiences at work and a positive, fulfilling, cognitive-affective state of mind and motivation at work [15]. According to Bakker [16], it is a mental state in which a working person is “fully immersed in the activity, feeling full of energy and enthusiasm about the work”. As a concept, work engagement is defined as a positive, fulfilling, work-related state of mind, and it is divided into three positive dimensions: vigour, dedication, and absorption [17]. Vigour encompasses the experience of high energy, mental resilience, persistence and willingness to invest effort at work. Dedication includes, for example, experiences of enthusiasm, involvement, inspiration and pride as well as sense of significance and meaningfulness at work, and absorption is characterized by concentration, dedication and enjoyment from work [16,18].

Research has shown that it is easier for committed and dedicated employees who consider their work to be meaningful to cope with adversity [19]. Work engagement and its positive effects can also be reflected in private life [18,20]. According to Hakanen [18], women experience work engagement slightly more often than men, and the elderly more often than young people. A later study also found that work engagement was higher in female than in male teachers [21]. The connection between work engagement and stress is interesting, since there is evidence that female teachers have higher stress than male teachers, while male teachers’ job satisfaction is higher than that of their female colleagues [22,23].

The term ‘**burnout**’ has been used to characterize a psychological state caused by prolonged job-related stress. Burnout has been divided into three independent dimensions: exhaustion, cynicism and inefficacy [10], which have different causes and consequences [24]. The (emotional) exhaustion is described as being emotionally depleted and distraught. Cynicism (or depersonalization) can be seen as mental distancing from exhaustive and discouraging work. Inefficacy (or low personal accomplishment) indicates reduced personal and/or professional efficacy caused by chronic stress [25]. (Without going into details about development and validation of the survey (for research on that, see, e.g., [26]) we refer to the 4th edition of the survey [25]). According to [24], exhaustion is strongly related to job demands, whereas cynicism and inefficacy are related to a lack of job resources. Even though the three dimensions are independent, they are interconnected: as exhaustion increases and empathy decreases, one’s sense of inefficacy also increases. An earlier study has also shown that burnout is negatively correlated with work experience (see [24] for a summary) and positively correlated with job dissatisfaction [27].

Taken together, it seems important to differentiate between the experience of work engagement and the experiences of workload and burnout. Work engagement can occur in spite of workload, and it can be a positive way to concentrate on work regardless of workload and negative psychological conditions. Conditions at home are important as well, since they can either balance the effects of work or make them even more stressful.

The concepts of **positive psychological well-being or mental well-being** include, for example, mental resources; abilities, opportunities and strengths; possibilities to influence one’s life; hopefulness and optimism; existence of satisfactory social relations or social capabilities; positive perception of oneself and one’s opportunities for development; positive emotions; and resilience. Positive mental well-being is a resource that can and should be developed and utilized, and it is a state of mind rather than absence of mental disorder [7,28]. Keyes states that positive psychological well-being can protect a person from somatic diseases, and it has also been shown to protect a person from suicide and learning difficulties [29]. Work-related mental well-being is closely connected with work engagement [28].

**Mental health** has been described as a diverse collection of non-specific symptoms of stress, anxiety and depression. Work is generally good for mental health, provided that the working conditions are convenient and the work itself satisfactory. Further, various issues in one’s personal life are important in either preventing or increasing mental health problems (for a summary, see [30]).

Studies have shown that there are **gender differences** in experiences of health and well-being at work. Hammarström, Lundman & Nordberg [9] used Connell’s relational theory of order [31] and stated that gendered experiences of health can be grouped into four categories, and three of them (becoming someone, being appreciated at work and having control over work) were work-related, while one (having good family relationships) was home-related. Gender differences in experienced workload and work engagement have also been explained by combining work engagement theory with gendered organization theory [32]. The research showed that women might not have the same opportunity to be engaged in the workplace as men because of organizational structures and practices, worker agencies and cultural assumptions; it is easier for men to experience psychological meaningfulness, safety and availability at work, and these three psychological conditions lead to work engagement. Especially women who have family responsibilities and who have to overcome barriers present in the gendered organizations need to make extra effort to demonstrate work engagement [32]. Finally, a recent study [30] with workers in various professions showed that females tend to report higher psychological distress than males, but for different reasons [30].

## 2. Materials and Methods

### 2.1. Data Collection

The participants were contacted via the church musicians’ trade union (Suomen kanttori-urkuriliitto) that informed all its members about the study by e-mail and sent the link to the questionnaire via the same e-mail. The church musicians were also informed about the research in social media (e.g., Facebook) in order to also recruit those who possibly had not read their e-mail or had ignored it. We received 99 responses. The number of union members still at work was 632, indicating that the response rate was 15.7%. Even though the response rate was low, the respondents were reasonably evenly distributed in the nine dioceses of Finland. The age profile was also close to that of all church musicians. Among the respondents, none identified themselves as non-binary/other, and seven (7.1% of the sample) did not disclose their gender. It is possible that at least some of those who did not want to disclose their gender were non-binary, but since this couldn’t be determined, we excluded them from the sample. The size of the group would, in any case, have been too small to analyze. The final sample (92) included 22 males (23.9%; response rate, 10.6% of male union members) and 70 females (76.1%; response rate, 17.3% of female union members). The higher female response rate is in line with earlier studies [33,34]. The mean age of the females was 48.8 years (SD 9.01) and that of the males was 52.3 (SD 9.71) years.

### 2.2. Variables and Standardized Measures

As stated above, we collected both quantitative and qualitative data (mixed-methods approach) to obtain a thorough understanding of the participants’ experiences of health and work-related well-being. The questionnaire included, first, questions about the participants’ lifestyle, e.g., about the possibility of eating lunch during a work day, their sleep patterns, alcohol use, smoking and weight. To measure alcohol consumption, we used the 3-question AUDIT-C screen [35]. We also assessed the possibility for participants to combine work and family responsibilities through responses to the following seven statements: (1) When I come home from work, I stop thinking about my work; (2) When I am at work, I do not think about home issues; (3) I can easily differentiate between work and home even when I telecommute at home; (4) I feel that I am neglecting domestic issues because of my work; (5) I sometimes neglect my family when I am wholly absorbed in my work; (6) I often find it difficult to concentrate on my work because of domestic issues; and (7) I have more energy to be with children when I also go to work. The participants responded on a four-step scale: 1 = “yes, absolutely”, 2 = “yes, mostly”, 3 = “mostly not”, and 4 = “absolutely not”.

The questionnaire also collected data by standardized measures. Work engagement was studied with the 9-question version of the Utrecht Work Engagement Scale (UWES) [17,36]. Each of the three sub-scales included three questions. Vigour was assessed with, e.g., “At my work, I feel bursting with energy”; dedication with, e.g., “I am enthusiastic about my work”; and absorption with, e.g., “I am immersed in my work”. Internal consistency was high in our data: Cronbach’s alpha for vigour was *α* = 0.927, for dedication *α* = 0.930 and for absorption *α* = 0.887. The 16-item Maslach Burnout Inventory (General Survey, MBI-GS) [25] was used to measure burnout experiences. The measure has 16 questions that form three subscales of burnout: exhaustion, cynicism, and professional efficacy. Both exhaustion (e.g., “I feel emotionally drained from my work”) and cynicism (e.g., “I doubt the significance of my work”) were assessed with five questions, and efficacy was assessed with six questions, e.g., “In my opinion I am good at my work”. All responses were given on a seven-point rating scale ranging from 1 (‘never’) to 7 (‘daily’). The efficacy was coded as inefficacy (lack of professional efficacy) to have the scales in the same order. Cronbach’s alpha for exhaustion was *α* = 0.934, for cynicism *α* = 0.891 and for inefficacy *α* = 0.844, all higher than in [20] and showing high internal consistency. Positive psychological well-being was measured using the seven-question shorter version of the Warwick-Edinburgh Mental Well-being Scale (SWEMWBS) [37]. The measure includes questions such as “I have been optimistic about the future” and “I have been thinking clearly”. The participants evaluated their experiences on a 5-step scale (1 = always; 2 = often; 3 = now and then; 4 = seldom, 5 = never). Cronbach’s alpha for SWEMWBS was *α* = 0.841, which was high and close to that obtained in [37]. Finally, the 5-item Mental Health Inventory (MHI-5) was used to measure psychological distress [30,38,39]. It consists of five questions, two of which are related to positive mental health and three to negative mental health. The questions are as follows. “How much of the time during the last four weeks have you: (1) been a very nervous person?; (2) felt downhearted and blue?; (3) felt calm and peaceful?; (4) felt so down in the dumps that nothing could cheer you up?; and (5) been a happy person?”. We recoded all the questions on the same-order scale so that the lowest responses on the five-point scale (1 = always; 2 = often; 3 = now and then; 4 = seldom, 5 = never) indicated that the participants had low mental health. Cronbach’s alpha for MHI-5 was *α* = 0.885; which was high and very close to that obtained in [30].

Further, the questionnaire included open-ended questions concerning experienced health of the participants (diseases and symptoms, physical condition, physical exercise), and their experienced ability to act, as well as the participants’ experiences of work. They all were directly connected with the previous question; e.g., after the question “Has anybody mistreated you or behaved inappropriately or menacingly?”, there was an option to tell more, and after asking “How has the COVID-19 pandemic affected you work?”, we asked the participants to tell more about the changes. Hence, the open-ended questions were of the type “Please tell more”, “Tell more if you like” or “Briefly describe”. These questions provided us with voluntary responses of varying length. It was also possible for the participants to leave the response boxes empty.

### 2.3. Data Analysis

The data consisted of the responses to the above-mentioned questions, standardized measures, and responses to open-ended questions. The quantitative data were analyzed mainly by statistical methods. As stated, there was low number of male participants and there was clear gender imbalance among the participants. To make the comparison of the participant groups more reliable and to address the possible bias caused by systematic differences in using the scales, we applied the following procedure with the standardized measures UWES MBI-GS and SWEMWBS, as well as the ‘combining work and home responsibilities’ question. We first summed the responses over the dimensions of the measure and then categorized the church musicians into three groups (low, average and high) by using average and ±1 standard deviation as cut-off points. The females and males were then compared by chi-square tests. With the MHI-5, we used the same cut-off point (52) as used in other studies e.g., [30], and for the body mass index (BMI), we used the cut-off points 25 and 30. In some analyses (e.g., the amount of sleep), we used the independent samples t-test. The cut-off points of the other measures are described in the Results section below.

The qualitative data (open-ended questions) were analyzed using a theory-driven analysis. Individual comments (single words or short expressions) concerning any issue that was related to the topic of the research (e.g., health or illnesses, stress, burnout, work load, working times, duties, ergonomics, management, harassment, family, congregation, sleep, eating, exercising, pandemic) were first extracted and classified; new classes were added as new content was found. During this process we extracted comments regardless of the original question; for example, whenever the participants wrote something about their ergonomics, the comment was classified accordingly regardless of whether it was a response to the question asking about their health or about changes caused by COVID-19. The classes were then grouped according to the research questions and with the help of the theory as follows: Category 1 included comments on health; Category 2 included comments on workload, distress and burnout, and Category 3 included comments that gave reasons for the experiences in Category 2. Comments containing self-criticism, which were written solely by female participants, were collected outside the theoretical framework. They were included in Category 1 because they aways related to either eating, being overweight or exercising. Finally, we analyzed pandemic-related positive experiences that formed Category 4. Figure 1 shows the study in outline.

## 3. Results

We report, first, the quantitative data, and after that the responses to the open-ended questions. We begin the quantitative data analysis with health-related issues, and then report the responses to the standardized measures. By the analysis of the open-ended questions we want to deepen the understanding of the distresses and to explain the responses to the measures.

### 3.1. Results from Quantitative Data

More than 80% of the church musicians had time to **eat lunch** during work days often (every day or weekly), and 10% had time for lunch only once a month or even more seldom (Figure 2). The female church musicians **slept** 7 h and 32 min per night, whereas the males slept less (6 h and 47 min), and the independent samples t-test showed that the difference was statistically significant (t_89_ = 2.93, *p* = 0.002). For females, the duration of sleep was close to the average of females in Finland (7 h and 24 min), whereas the males slept 31 min less than the average male in Finland (7 h and 18 min) [7]. More than half of the church musicians reported that they often sleep enough, and the share of females feeling so was higher (68.2%) than the share of males (55.0%), yet the chi-square test showed that the difference was not statistically significant (*p* = 0.088). The share of both males and females who felt that they sleep enough was lower for church musicians than for Finns in general (females: 74.6%, and males: 78.0%) [7].

As stated, **alcohol use** at risk level was measured using the 3-question AUDIT-C screen, in which each question was scored between 0 and 4. If the sum of the scales, which in theory can be between 0 and 12, is equal to or higher than 5 for females and 6 for males, there is an increased risk of alcohol-related problems [40]. Of the female church musicians, 8.6% were at or exceeded the 5-point limit, while the percentage of male church musicians who were at or exceeded the 6-point limit was higher (18.2%), but the difference was not statistically significant (χ^2^ = 1.60, *p* = 0.207). Both male and female church musicians were, however, far below the averages (16% for females and 28% for males) found for a general population sample in Finland 2016 [41]. Further, the percentage of those who did not use alcohol at all was high both for females (28.6%) and males (31.8%). In the Finnish population the share of those who do not use alcohol at all is 20.3% of females and 15.7% of males [7].

Altogether, 14.3% of male church musicians and none of females **smoked or used snuff** daily (for the general population the figures are 15.6% of males and 10.7% of females) [7]. The majority of respondents did not smoke nor take snuff at all (87.1% of females and 57.1% of males), had stopped it (7.1% and 19.0%, respectively) or smoked/used snuff occasionally (5.7% and 9.5%, respectively).

For those 63 females and 21 males who responded to the height and weight questions, 34.9% of females and 28.6% of males had **body mass index** that was normal (18.5 ≤ BMI < 25). Approximately one third (28.6% of females and 38.1% of males) were marginally overweight (25 ≤ BMI < 30), and the rest (36.5% of females and 33.3% of males) fell within the obesity range (BMI ≥ 30). The share of church musicians with normal weight was approximately the same as in the Finnish population [7], but the share of church musicians with BMI ≥ 30 was higher for both males and females (Figure 3).

We analyzed the church musicians’ experiences about combining home and job responsibilities and separating home and work. Since some of the seven statements estimated positively the ability to combine work and home and some others estimated the neglect of either work or home, we first harmonized the direction of the scales. We then averaged the responses over the questions and categorized the responses into three groups: often (low score: lower than mean − 1 st. dev.), average (mean ± 1 st. dev.), and seldom (high score: higher than mean + 1 st. dev.; see Figure 4). The results showed that combining work and home responsibilities was difficult for approximately one fifth (17.1%) of the female church musicians and one tenth (9.1%) of male church musicians, but the responses were not gendered (χ^2^ = 0.847, *p* = 0.655; *n* = 92).

For the **Utrecht Work**
**Engagement Scale** (UWES), we summed the responses over the three dimensions and grouped the sum score into three categories as previously explained (mean ± 1 standard deviation as the cut-off points; see Figure 5). Generally, the responses showed a smaller share of high (positive) and a larger share of low (negative) responses for males than for females, but the difference was not statistically significant (χ^2^ = 1.802, *p* = 0.406; *n* = 92). The responses showed that more than one fourth (27.3%) of the male church musicians and one fifth (19.1%) of female church musicians had low experiences of work engagement.

The participants’ responses on the **Mental Health Inventory** (MHI-5) were summed and the sum scale modified to be from 0 to 100. Using the cut-off point 52, we determined the proportion of church musicians who had psychological distress on the level that is clinically significant and needs to be treated (see e.g., [30]). This analysis revealed the alarming result that more than one third (36.8%) of male and nearly one fourth (23.1%) of female church musicians were distressed, but the responses were not gendered (χ^2^ = 1.117, *p* = 0.291; see Figure 6). The figures were clearly higher than those of the general Finnish population (13.6% of males and 14.9% of females) [8].

On the shortened version of the **Warwick-Edinburgh Mental Well-Being Scale** (SWEMWBS), low scores indicate high mental well-being. The summed responses were grouped into three mental well-being categories (low, average and high) using the mean ± 1 st. dev. as previously described. The results were similar to the other measures reported above: most of the church musicians estimated their mental well-being as average, not high nor low (the grey portions of the bars in Figure 7), and there were no gender differences (χ^2^ = 1.356, *p* = 0.508). Approximately one tenth (9.1%) of male and one eighth (12.9%) of female church musicians had low mental well-being. The proportions were smaller than in the Finnish population, in which the share of those who have low mental well-being is 16% of females and 17% of males [7].

The responses on the three dimensions (exhaustion, cynicism and inefficacy) of the **Maslach Burnout Inventory General Survey** (MBI-GS) were averaged, and the averages were grouped into three categories (low, average and high) using the mean and standard deviation as described. The three categories are shown in Figure 8 with the red sections of the bars representing high scores (indicating burnout) and the green sections indicating low scores. Ahola et al. [42] found previously that females score higher on exhaustion, whereas males score higher on cynicism, but in our study the responses were not gendered. The statistics for the three dimensions were as follows: Exhaustion (χ^2^ = 0.075, *p* = 0.963), Cynicism (χ^2^ = 2.023, *p* = 0.364) and Inefficacy (χ^2^ = 2.679, *p* = 0.262). The most concerning result was that a little more than one-sixth of the female (18.6%) and a little less than one-sixth of the male (15.2%) church musicians had high scores, indicating that they often had burn-out symptoms. Compared to the results of Ahola et al., the means on all three scales were higher in our study, indicating more burnout among church musicians [42].

### 3.2. Results from Open-Ended Questions (Qualitative Data)

The above analyses did not reveal any differences between genders. They, however, revealed that, depending on the measure, from 9% to 37% of male and from 13% to 23% of female church musicians were not well, since they had low work engagement, low mental health, low mental well-being and burnout symptoms.

Since we wanted to analyze the working conditions more thoroughly, and especially experiences that can explain distress, we analyzed the data collected by open-ended questions. This analysis also focused on possible gender differences. As stated, we extracted individual comments (single words or short expressions) from the responses, collected them into classes and used theory-driven analysis to categorize them. Unlike the quantitative analyses, the analysis of these data revealed a clear difference between male and female church musicians: the females were more willing and capable to analyze and verbalize their work and their experiences than their male colleagues. This was the case even when the number of respondents was taken into account. Below, we describe the categories and provide some quotations (in italics) from the data. We also provide the number of comments in each category from female and male respondents as the absolute number (and in parentheses as a ratio of 100) to show differences between the groups (see Table 1).

The number and ratio of comments on **health** (Category 1) was 104 (149/100) from females and 18 (82/100) from males. There were names of diseases (e.g., *scoliosis*; *osteoarthritis*; *coeliac disease*; *asthma*; *hypertension*), descriptions of poor ergonomics (*sedentary work*; *carrying instruments*; *steep stairs*), as well as comments on too little exercise (*too little exercise*; *no strength training*). The church musicians also wrote about increases in weight and their unhealthy diet (e.g., *my diet is unhealthy*; *too large portions*; *candies nearly every day*; *too much meat*; *too much salt*). A peculiar aspect of this category was the female church musicians’ comments including strong self-criticism. The comments usually included two sentences joined by ‘but’; the first sentence indicating something positive and the second one adding that more should be done. *I exercise a lot but there is not enough strength training; I cook healthy home food but eat too much delicacies; Even though I exercise many times a week, in the back of my mind I think that I should exercise more; I exercise a lot and I have no health problems, but the fact is that I weigh too much in relation to my height.*

The number of comments **describing distress and burnout** (Category 2) was 95 (136/100) from females, and the ratio was more than three times higher than the ratio of comments from males (9 comments; 41/100). As stated in Section 2.1, females normally respond more often than males, but for the Category 2 responses, the difference was strikingly large. The comments describing distress and burnout tell a stark truth about female church musicians’ work experiences. The comments included direct words describing the phenomenon, such as *exhaustion*, *burnout*, *distress*, *depression* and *apathy*. *Insomnia* and *frustration* occurred frequently in the data and so did *feeling oneself worthless* or *insignificant*. Loneliness included both *working alone* (either at home or in the empty church) and the *absence of colleagues.* Further, *the meaning of work disappeared* when the church musicians were *not able to meet parish members*.

The **reasons for distress and burnout** (Category 3) were also included in the open-ended responses. In this category, the number and ratio of females’ responses was higher (169 comments; 241/100) than males’ (21 comments; 95/100), but the difference was smaller than in Category 2. As shown by the statistical analyses, a relatively equal proportion of female and male church musicians were unwell, and both groups were willing to elaborate the reasons. Category 3 included, first, comments about the ambiguity of job descriptions, telling that there was *too much work*, *extra work*, *haste*, *work without working hours* and *long working days*. Further, the church musicians did not know *what duties were part of their work*; they told about *scattered work* with *unrealistic expectations* and *constant increase of new work*. The most often repeated new work causing stress was *live streams*, and it is notable that most of these comments (14 out of the total of 16) came from female church musicians. The streaming itself was a challenge (*pressure caused by live streams; live streams continue to be accompanied by ongoing inappropriate criticism and side directing*). Live streams also caused extra work (*huge amount of extra work on the internet; preparing for work tasks takes more time* [because of live streams]), and the church musicians felt that they were given no education nor support with the equipment and software that was new for most of them (*new things*, e.g., *streaming and video making, I have had to learn very quickly*; *the technical side of the live streams fell to the church musicians*; *no support from the superior*).

The second group of comments included managerial problems such as poor management, a lack of leadership and leaders who do not know the everyday reality. Managerial problems appeared also as a lack of support from superiors, inadequate communication and instructions as well as a poor workplace atmosphere. Further, the consequences of poor management could be seen as constant changes, difficulties in planning work and the impossibility of keeping to plans, and these all caused stress in both male and female church musicians. The comments about changes and interruptions were grouped separately, even though they could be seen as an aspect of the job description ambiguity or as managerial problems. All 32 comments came from the female church musicians, and the main content was that there is much planning but no guarantee that the plans can be carried out; thus, one should always have a plan B or be ready to improvise.

Comments concerning inappropriate conduct at work constituted the fourth group of comments. As with the previous comments, the number of these comments was also higher from females (44 comments; 63/100) than from males (5 comments; 23/100). The comments included *mistreatment* at work, *bullying*, *mischief*, *understating* and even *threats and intimidation* from both colleagues and superiors. *Inappropriate comments* could come from the members of the work community and sometimes also from the parish members. Both *sexist talk* and *sexual harassment* were mentioned twice by females.

The open-ended responses included also **positive experiences** concerning the effects of the pandemic on the work of the church musicians (Category 4). As with earlier comments, the females wrote more comments (58 comments; 83/100) than males (6 comments; 27/100). The most frequently occurring comments included the *decrease in the amount and fragmentation of work* as well as the *decrease in distress*. The church musicians could use *more time for exercising*, they *slept better*, they *felt better*, had *more time with the family*, and they had *time for practicing* and *creative processes*. As one church musician put it, *work during the corona was exactly what I had dreamed of all my life, and why in time I went to study. Me alone in the organ loft!*

## 4. Discussions

Our study revealed that some lifestyle aspects of the church musicians were on a better level than in the general population: the church musicians smoke and use alcohol less than the average population. It also showed that during the last year the experiences of church musicians’ workload, burnout, mental health and engagement in work were mostly between the extreme ends of the continuum. Regardless of the measure we used in the analyses, there were always from 15 to 20% of church musicians who felt well, and we can conclude that they had personal characteristics and resources as well as behavioural strategies [19] that were strong and helped them manage their demanding working conditions in general and the changes caused by the pandemic in particular. However, depending on the measure, there were also between 9% and 37% of church musicians who were not well, and very often, there were more of those feeling unwell among church musicians than in the general population [7,8]. The proportion of church musicians who had high mental distress (37% of males and 23% of females) and burnout symptoms (15.2% of males and 18.6% of females) was especially alarming. The open-ended responses further explained these results through comments about health problems and experiences of distress, work load and burnout. An earlier interview study [43] also revealed various stress factors (the impossibility to manage the fairly limitless set of tasks, fragmentation of work in terms of time and tasks and ambiguity of work boundaries) that were central in our study as well. Experiences of work are not constant every day but change according to events, job demands and job resources [16], and as stated, the work of church musicians includes much daily variation. In addition to these general stressors, our study revealed stressors related to the additional challenges caused by the COVID-19 pandemic, especially live-streams of the services.

Our basic question about the experiences of male and female church musicians was answered in our study, and the finding was that the experiences measured by the four standardized measures were not gendered, that is, we did not find differences in burnout, mental health and well-being. Unlike earlier studies [18,21], we did not find gender differences in work engagement. Neither did we find significant differences between male and female church musicians in eating, risky alcohol use, smoking or body mass index. Further, we did not find any difference between males and females in stress caused by combining work and home responsibilities, a finding that differs from earlier research results stating that males have high-level mental distress at work, whereas with females, the distress is related to the family-work imbalance [30].

The results from the quantitative and qualitative data were consistent for the male church musicians but not for the females. For the latter group, the qualitative data revealed distress, work load and burnout. Our study did not reveal reasons for the contrast, that is, we cannot say why female participants did not want to choose the ‘daily’ or ‘very often’ option in the direct questions about burnout or psychological distress, even though their experiences of distress were numerous, as the qualitative data showed. This finding, however, shows the importance of a mixed-methods approach, since the qualitative data revealed stressors that could not be identified by the quantitative measures.

The results are subject to several limitations. The number of respondents and the response rate was low, especially for male church musicians. Because of the small number of participants, the analyses did not show statistical significance, even though with some measures, there seemed to be a difference between male and female church musicians. Neither were we able to analyze the effect of age on the responses. Further, we did not in any way control the responses to the open-ended questions, and the activity of the females could be seen in those responses as well. However, we believe that the use of mixed methods compensated for the limitations caused by the small sample size, and the qualitative data provided us with rich, detailed and sensitive data describing the working conditions and the stressors of church musicians, something that would not have been possible to collect by using standardized measures only.

## 5. Conclusions

As a whole, the data showed that 37% of male church musicians and 23% of females had severe stress. These experiences were explained in the open-ended questions by the females: they reported clearly more often about their negative experiences than their male colleagues. Unlike the quantitative measures, the open-ended responses were in line with an earlier study showing that females report about their psychological workload more often than males [30]. Stress can be related to expectations [30], and the responses mentioned *unrealistic expectations* but did not reveal whose expectations should have been met, whether the female church musicians’ own or those of others. However, there remain questions about why only females told about their experiences in the open-ended questions and why their stress was not reported in the quantitative measures in our study. We ask two hypothetical questions to direct future research, the first of which concerns recognition of stress: Is it possible that the female church musicians did not respond that they were unwell and felt stressed ‘daily’ or ‘all the time’ because they—at the average age of 50 and after having worked as church musicians for more than 21 years—were so used to the stress and to the bad feelings that it had become a normal state at work?

Research on work engagement and work-home balance has shown that work engagement is positively correlated with working overtime, which also can mean taking work home. Research has also shown that those who invest a lot in work without receiving the appropriate appreciation are exposed to burnout (for a summary see [32]). In our study, only the female church musicians mentioned underestimation and disparagement in the open-ended responses, even though this was not seen in the standardized measures. From this follows our second hypothetical question: Is it possible that the female church musicians are so used to the challenges in combining home and work that they do not even recognize that it causes stress? The interpretation is in line with Connell’s relational theory of order [31], especially with the component ‘being appreciated at work’. The interpretation is, further, in line with the idea that females need to expend extra effort to demonstrate work engagement [32] and that they hit the glass ceiling [5] (pp. 83–88).

As stated, the starting point for our study was the finding that female church musicians’ SMRs for cancers and cardiovascular diseases were higher than those of males. The results of the study could not explain earlier deaths, but analyzing qualitative data can reveal connections between stressors and causes of death; for example, one meta-analysis revealed a statistically significant connection between job-related distress and hypertension [44]. Another meta-analysis revealed that perceived psychosocial stress was associated with increased risk of stroke, and the risk was higher for females than males [45]. Further, stress and poor mental health of young adults has been shown to be associated with poor physical health, including, e.g., substance abuse disorders, smoking, minimal physical activity and neglect of nutritional guidelines, and poor health behaviours, which can lead to greater risk for cancers and other chronic health conditions [46]. Workers who have continuous work stress might not have enough time and resources to take care of their physical health. Especially stress that becomes a “normal state” can go unnoticed, and no alleviation can be found.

Because of the contradiction between the measures and the open-ended responses and because of the hypothetical questions raised by the results, we call for further research on work-related well-being of females. We also suggest mixed-methods approaches for collecting data in studies examining gender differences in appreciation and control at work as well as in connections between home and work.

## Figures and Tables

**Figure 1 ijerph-19-09866-f001:**
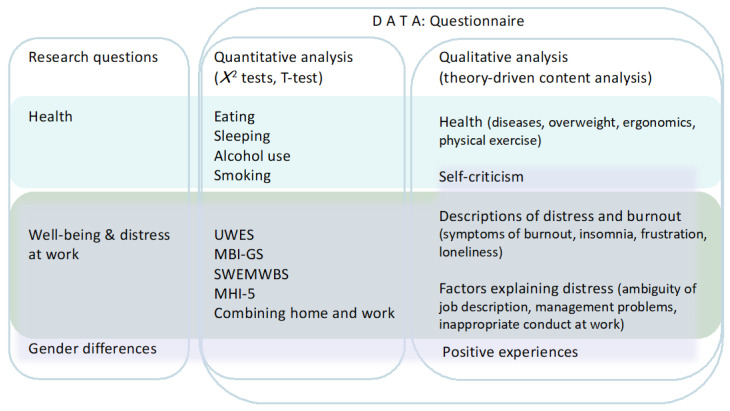
The study in outline: research questions, data, measures and analyses.

**Figure 2 ijerph-19-09866-f002:**
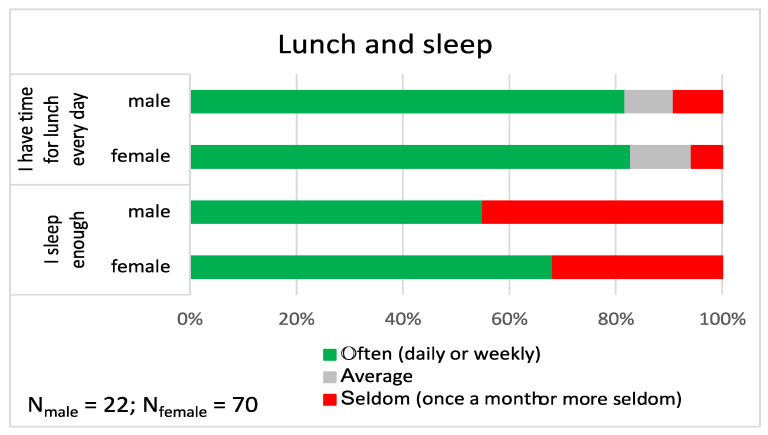
Lunch and sleep patterns of the participants. The share of participants with “often”, “average” and “seldom” responses are shown separately for male and female church musicians (in percentages).

**Figure 3 ijerph-19-09866-f003:**
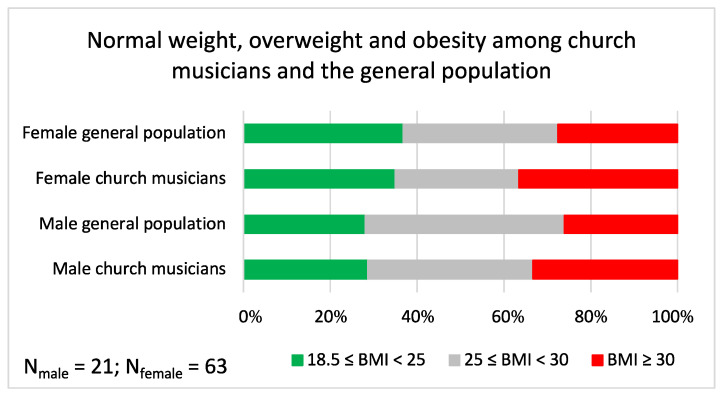
Proportions of respondents with normal weight, overweight and obesity among church musicians of the study and the Finnish general population (data shown as percentages).

**Figure 4 ijerph-19-09866-f004:**
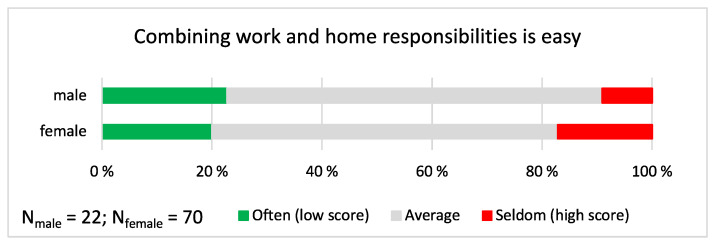
Combining work and home responsibilities. The proportion of “often”, “average” and “seldom” responses are shown separately for male and female church musicians. On the scale between 1 and 4, the mean was 2.61, and the mean ± 1 standard deviation cut-off points were as follows: Often ≤ 2.167; Average 2.168–3.053; Seldom ≥ 3.055.

**Figure 5 ijerph-19-09866-f005:**
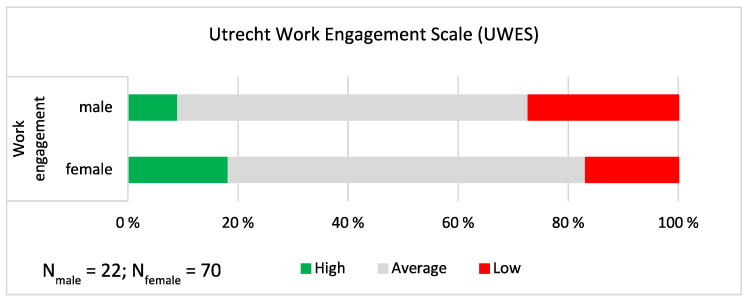
Work engagement. The proportions of “high”, “average” and “low” responses shown separately for male and female church musicians. On the sum scale between 1 and 21, the mean was 14.852, and the mean ± 1 standard deviation cut-off points were as follows: Low ≤ 10.538; Average 10.539–19.165; High ≥ 19.166.

**Figure 6 ijerph-19-09866-f006:**
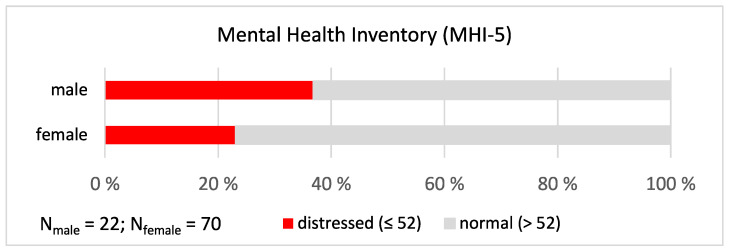
Mental Health Inventory. The proportions of distressed male and female church musicians.

**Figure 7 ijerph-19-09866-f007:**
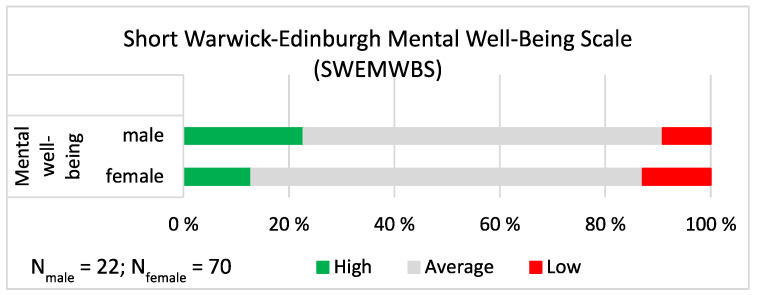
SWEMWBS. The share of high, average and low mental well-being shown separately for male and female church musicians. On the sum scale between 7 and 35, the mean ± 1 standard deviation cut-off points for mental well-being were as follows: High ≤ 12.811; Average 12.812–20.840; Low ≥ 20.841.

**Figure 8 ijerph-19-09866-f008:**
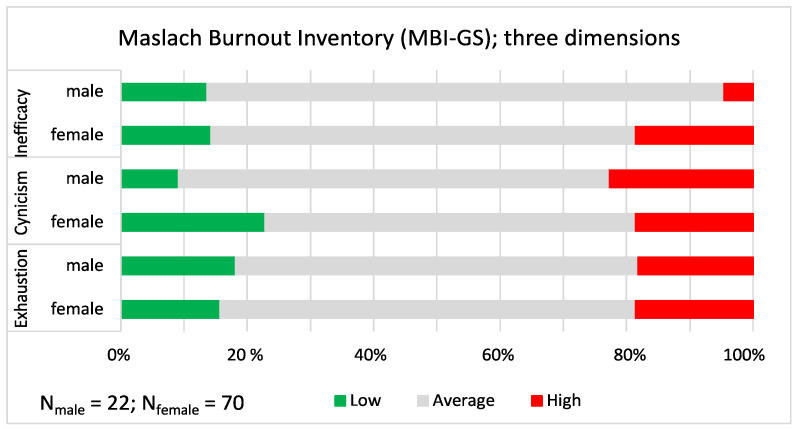
Maslach Burnout Inventory. The proportions of low, average and high burnout on three dimensions shown separately for male and female church musicians. On the average scales between 1 and 7, the mean and the mean ± 1 standard deviation cut-off points were as follows: Exhaustion; mean = 3.298; low ≤ 1.558; average 1.559–4.884; high ≥ 4.885; Cynicism; mean = 3.439; low ≤ 1.777; average 1.778–5.100; high ≥ 5.101; Inefficacy; mean = 2.993; low ≤ 1.812; average 1.813–4.172; high ≥ 4.173.

**Table 1 ijerph-19-09866-t001:** Main categories, subcategories, the number of responses and the share of 100, shown separately for male and female church musicians.

Main Category	Subcategories	Number of Responses from 70 Females (Share/100)	Number of Responses from 22 Males (Share/100)
Health		104 (149/100)	18 (82/100)
	Illnesses and symptoms	35	8
	Weight and eating	40	6
	Ergonomics	16	1
	Training and exercising	13	3
Description of stress and burnout		95 (136/100)	9 (41/100)
	Burnout and distress	79	5
	Loneliness	10	4
	Other	6	0
Reasons for distress and burnout		169 (241/100)	21 (95/100)
	Ambiguity of work description	45	8
	Poor management	48	8
	Changes and interruptions	32	0
	Inappropriate conduct at work	44	5
Positive comments		58 (83/100)	5 (23/100)

## Data Availability

The data (without responses to the open-ended questions) are available on request from T.K. The original questionnaire is available on request from T.K.

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
