# Peer review of "Health and Well-Being of Church Musicians during the COVID-19 Pandemic—Experiences of Health and Work-Related Distress from Musicians of the Evangelical Lutheran Church in Finland"

_ijerph, 2022, doi:10.3390/ijerph19169866_

Round 1
Reviewer 1 Report
I am glad to have this opportunity to review this article again. After reading it, there are also some minor problems. The problems are listed as follow for your reference.
1. Why choose church musicians in Finland?
2 The data source of survey comes from the single source, which may cause CMV, but the article does not address it.
3. The research does not include control variable(s), which may affect the results.
4. In Page 10, MBI-GS scale is a seven-point rating scale. But in Figure 8, the scale is a three-point scale, why?
Author Response
Reviewer 1 Comments:
I am glad to have this opportunity to review this article again. After reading it, there are also some minor problems. The problems are listed as follow for your reference.
- Why choose church musicians in Finland?
Authors’ response: We tell about the earlier study concerning Finnish church musicians in the very beginning of the article, and the lack of gender comparison in the second paragraph. We tried to further clarify why we wanted to run this particular study (Introduction, 2nd paragraph)
- The data source of survey comes from the single source, which may cause CMV, but the article does not address it.
Authors’ response: Since the background study on gender differences in mortality was from Finland and from church musicians, we wanted to carry out our research in Finland among church musicians. As stated in 2.1 (please see changes there), the questionnaire was sent to all members of the Church Musicians’ union, and most church musicians are members of the union. We believe that the single source was not a problem.
- The research does not include control variable(s), which may affect the results.
Authors’ response: See above. We did not need to control for the participants.
- In Page 10, MBI-GS scale is a seven-point rating scale. But in Figure 8, the scale is a three-point scale, why?
Authors’ response: As described, we used average ± 1 standard deviation as cutting points for the measure regardless of the original rating scale. The chosen cutting points divide the responses into three groups. We added clarification in the text (section 2.3).
Reviewer 2 Report
This study aims to examine gender differences in health and work-related experiences of the church musicians of the Evangelical Lutheran Church of Finland in 2020–2021.
Although the objective seems clear, however, the introduction is not. Many ideas and concepts are introduced, but internal coherence is lacking to guide and connect the discourse.
The manuscript is relatively poor methodologically. For example, the reliabilities of the instruments used have not been considered. Further, since there are results from previous studies, working hypotheses could have been proposed to guide the analyses.
The graphs presented do not add too much to the content of the study. Since the selected cutoffs for the three categories are made relative to the mean and ± 1 SD, the graphs represent a normal distribution; that is, data follows a normal distribution.
In the discussion, some conclusions drawn are not supported by the results. For example, the authors note, "we can conclude that the personal characteristics and resources of those church musicians, together with their behavioral strategies, were strong and helped them manage their demanding working conditions also during the pandemic." This statement is not clearly supported by the results analyzed.
The qualitative analysis is not evident. The coding criteria are not clear. Furthermore, the results of those analyzes seem to contradict the results of the quantitative analyses. The authors point out that there are no differences between genders in the quantitatively analyzed variables (engagement, mental health, mental well-being, burnout). However, they point out that the qualitative analyzes reflect enormous gender differences.
The main problem is not the contradictory results but that there is no evidence of the authors' attempts to explain this difference.
In addition, the authors not only do not attempt to explain this difference (contradiction) in the results but, despite the evidence (of the quantitative results), insist on sustaining the existence of gender differences based on methodologically non-rigorous qualitative analyses.
In general, the manuscript is not coherent.
Apart from the data collected during 2020-2021, it is not clear how the research subject is related in any way to the pandemic situation, although this is mentioned in the title. The title should not include concepts that are not critical aspects of the study, as they confuse the reader.
Author Response
Reviewer 2 Comments:
This study aims to examine gender differences in health and work-related experiences of the church musicians of the Evangelical Lutheran Church of Finland in 2020–2021.
Although the objective seems clear, however, the introduction is not. Many ideas and concepts are introduced, but internal coherence is lacking to guide and connect the discourse.
Authors’ response: We have tried to clarify the introduction by shortening it, by explaining the structure and using bold print for the concepts.
The manuscript is relatively poor methodologically. For example, the reliabilities of the instruments used have not been considered.
Authors’ response: We had the understanding that the standardized measures are so often used that we do not need to explain them in the article. For example, the validity of the burnout measure (MBI-GS) has been cross-nationally and across occupations studied soon after it was introduced in 1996 (for summary, see Richardsen & Martinussen, Factorial Validity and Consistency of the MBI-GS Across Occupational Groups in Norway. International Journal of Stress Management 12(3), 289–297 from year 2005). We can add references to the measures, but this is something we would like the action editor to decide, since we are not quite certain about how familiar the measures are for the general reader of the IJERPH.
Further, since there are results from previous studies, working hypotheses could have been proposed to guide the analyses.
Authors’ response: Thank you for the suggestion. This is true. We could have defined a hypothesis for the quantitative analyses (in which such hypotheses can be either accepted or rejected), but not for the qualitative analyses, we decided not to write about hypotheses.
The graphs presented do not add too much to the content of the study.
Authors’ response: We agree that many of the percentages shown in the graphs are also written in the text (but not all). This is an issue about which we would like to discuss with the action editor: we believe that there are different kinds of readers, and figures serve those who want to get an understanding of the results at a glance.
Since the selected cutoffs for the three categories are made relative to the mean and ± 1 SD, the graphs represent a normal distribution; that is, data follows a normal distribution.
Authors’ response: The Reviewer has right about the basic idea that mean ± 1 SD is one defines normal distribution for its part. In our case, the cutting points indicate that approximately 68% of the whole set of data (males and females together) were within the range of mean ± 1 SD. Since the focus was in gender differences, the cutting points were used for grouping male and female respondents separately, and as such they show the differences between males and females in a reliable way by telling the share of females and males who were below or above the cutting points.
In the discussion, some conclusions drawn are not supported by the results. For example, the authors note, "we can conclude that the personal characteristics and resources of those church musicians, together with their behavioral strategies, were strong and helped them manage their demanding working conditions also during the pandemic." This statement is not clearly supported by the results analyzed.
Authors’ response: The whole sentence was “Approximately 15–20% of the church musicians felt well, and we can conclude that the personal characteristics and resources of those church musicians together with their behavioural strategies [19], were strong and helped them manage their demanding working condition also during the pandemic.”; the latter part refers to those 15-20% mentioned in the first part of the sentence. But since the sentence was not clear and could be misunderstood, we rewrote it to make the idea of some well-being church musicians more clear. (Discussion, 1st paragraph)
The qualitative analysis is not evident. The coding criteria are not clear.
Authors’ response: Thank you for the comment. We have added description of the coding (2.3, 2nd paragraph; 3.2, 2nd paragraph)
Furthermore, the results of those analyzes seem to contradict the results of the quantitative analyses. The authors point out that there are no differences between genders in the quantitatively analyzed variables (engagement, mental health, mental well-being, burnout). However, they point out that the qualitative analyzes reflect enormous gender differences. The main problem is not the contradictory results but that there is no evidence of the authors' attempts to explain this difference.
In addition, the authors not only do not attempt to explain this difference (contradiction) in the results but, despite the evidence (of the quantitative results), insist on sustaining the existence of gender differences based on methodologically non-rigorous qualitative analyses.
Authors’ response: We understand that quantitative analyses are often considered as more reliable than qualitative analyses, and we know that there is research in which only quantitative data can be used. We, however, believe that in our study the qualitative data and analysis is as important as the quantitative one, and that the mixed-method approach is fruitful for our research. We write about the contradiction showing that female participants responded differently in the quantitative and the qualitative part of the study. We cannot leave the results of the qualitative part unnoticed and write that the gender differences revealed by it do not exist. Perhaps the most important result of our work is that the quantitative measures might not tell the whole truth about female participants well-being. We have tried to clarify the contradiction of the findings (3.2, 2nd paragraph; Discussion, 3rd paragraph; Conclusions, last paragraph).
In general, the manuscript is not coherent.
Authors’ response: We hope that the changes made have improved the coherence of the article.
Apart from the data collected during 2020-2021, it is not clear how the research subject is related in any way to the pandemic situation, although this is mentioned in the title. The title should not include concepts that are not critical aspects of the study, as they confuse the reader.
Authors’ response: Thank you for the comment. In Introduction, at the end of the second paragraph we tell how the pandemic changed the church musicians’ work. We have added discussion about the pandemic situation also later in the article (2.3, 2nd paragraph; Results, last paragraph; Discussion 1st paragraph)
Reviewer 3 Report
Dear authors,
The authors conducted a study to explore
gender differences in health and work-related experiences of the church musicians with special focus on experienced stress and burnout and work engagement and mental well-being on the other. The data were collected by a questionnaire with standardized measures and open-ended questions.
Despite the positive features of the study, there are some considerations to take care before the paper is published.
The following comments will summarize my appreciation and major concerns with your paper. I hope these comments help you further improve your study.
Introduction:
1. The introduction is confuse and longer. It would be useful that the authors summarize the introduction and make a better connection between constructs.
Theoretical background
2. What are the main goals of the paper? What do you want to achieve? And what do you add to the literature.
Method
3. Please add more information regarding the data collection procedure. How did the participants were recruited?
4. Did the authors used any control variables?
Discussion
The discussion was short and did little more than summarise your findings.
5. Please, develop the discussion section, considering the theoretical implications.
6. The limitations and future research should be elaborated.
7. What are the main practical implications of the study? And what do you add that is not tested before?
Author Response
Reviewer 3 Comments:
Dear authors,
The authors conducted a study to explore
gender differences in health and work-related experiences of the church musicians with special focus on experienced stress and burnout and work engagement and mental well-being on the other. The data were collected by a questionnaire with standardized measures and open-ended questions.
Despite the positive features of the study, there are some considerations to take care before the paper is published.
The following comments will summarize my appreciation and major concerns with your paper. I hope these comments help you further improve your study.
Introduction:
- The introduction is confuse and longer. It would be useful that the authors summarize the introduction and make a better connection between constructs.
Authors’ response: We have tried to clarify the introduction by shortening it, clarifying the structure and taking care that we always write about work-related concepts.
Theoretical background
- What are the main goals of the paper? What do you want to achieve? And what do you add to the literature.
Authors’ response: We have tried to clarify the aims; see paragraph 2 of Introduction.
Method
- Please add more information regarding the data collection procedure. How did the participants were recruited?
Authors’ response: we added text about the recruitment process (see 2.1)
- Did the authors used any control variables?
Authors’ response: The questionnaire was sent to all members of the Church Musicians’ union, and most church musicians are members of the union. For this reason we did not need to control for the participants.
Discussion
The discussion was short and did little more than summarise your findings.
- Please, develop the discussion section, considering the theoretical implications.
Authors’ response: We have elaborated the discussion section and added references to earlier study
- The limitations and future research should be elaborated.
Authors’ response: We have rewritten the limitations and added ideas for future research
- What are the main practical implications of the study? And what do you add that is not tested before?
Authors’ response: We tried to improve the way we describe the main practical implication, that is, the need of more sensitive measures of studying well-being of female workers (Discussion, 3rd paragraph, Conclusions, last paragraph).
Round 2
Reviewer 2 Report
I have had the opportunity to revise this manuscript a second time, finding that this new version of the document has improved. The introduction has been shortened a bit and is more precise.
However, I’m still concerned about the Materials and Methods section.
Variables and standardized measures: This section of the manuscript remains general. More information is needed on the instruments used. Not only say what instruments have been used but what dimensions have been included and how their treatment has been. For example, engagement includes dedication, vigor, and absorption. How have these three dimensions been treated? What about the three burnout dimensions? The authors explain some of these points in the results section, but they should be included here and leave the results section just for reporting results.
Reliabilities remain unreported. Regardless of whether some instruments are considered standardized measures (i.e., UWES or MBI), the internal consistency of an instrument may vary according to the sample.
The authors still do not explain why to categorize measures that can be treated as continuous. Categorizing the measures is fine, but an explanation should be given as to why it should be done and what is the convenience or benefit for the research, knowing that when categorizing a continuous measure, information is lost.
The analysis for the qualitative data remains unclear to me and needs a more detailed explanation. What were the open-ended questions? Any criteria to select them? The authors mention that “the classes were then grouped according to the research questions.” What are the research questions? The authors note that comments of self-criticism were included in Category 1. These comments were only made by female participants. How might this affect Category 1 health comments made by female and male participants?
In summary, the methods section remains unclear and, in its current state, would not allow the study to be replicated.
The authors should comment on how the gender imbalance (24% men and 76% women) might affect the results.
Reviewer 3 Report
The authors presented an improved version of their study and addressed all the comments identified before. I wish them the best of luck with their research.
Author Response
We thank R3.